# Active search strategies, clinicoimmunobiological determinants and training for implementation research confirm hidden endemic leprosy in inner São Paulo, Brazil

**Fred Bernardes Filho**[1,2ʘ], **Claudia Maria Lincoln Silva**[1,2‡], **Glauber Voltan**[1,2‡], **Marcel Nani Leite**[1,2‡], **Ana Laura Rosifini Alves Rezende**[1,2‡], **Natália Aparecida de Paula**[1,2ʘ], **Josafá Gonçalves Barreto**[3‡], **Norma Tiraboschi Foss**[1,2‡], **Marco Andrey Cipriani Frade**[1,2ʘ] *

1 Dermatology Division, Department of Medical Clinics, Ribeirão Preto Medical School, University of São Paulo, Ribeirão Preto, Brazil, 2 Center of National Reference in Sanitary Dermatology focusing on Leprosy of Ribeirão Preto Clinical Hospital, Ribeirão Preto, Brazil, 3 Spatial Epidemiology Laboratory, Federal University of Pará, Castanhal, Brazil

ʘ These authors contributed equally to this work.
‡ CMLS, GV, MNL, ALRAR, GB, and NTF also contributed equally to this work.
* mandrey@fmrp.usp.br

**Data Availability Statement:** All relevant data are within the manuscript.

## Abstract

### Background

This study evaluates implementation strategies for leprosy diagnosis based on responses to a Leprosy Suspicion Questionnaire (LSQ), and analyzes immunoepidemiological aspects and follow-up of individuals living in a presumptively nonendemic area in Brazil.

### Methodology/Principal findings

Quasi-experimental study based on LSQ throughout Jardinópolis town by community health agents, theoretical-practical trainings for primary care teams, dermatoneurological examination, anti-PGL-I serology, RLEP-PCR, and spatial epidemiology. A Leprosy Group (LG, n = 64) and Non-Leprosy Group (NLG, n = 415) were established. Overall, 3,241 LSQs were distributed; 1,054 (32.5%) LSQ were positive for signs/symptoms (LSQ+). Among LSQ+ respondents, Q2-Tingling (pricking)? (11.8%); Q4-Spots on the skin? (11.7%); Q7-Pain in the nerves? (11.6%); Q1-Numbness in your hands and/or feet? (10.7%) and Q8-Swelling of hands and feet? (8.5%) were most frequently reported symptoms. We evaluated 479 (14.8%) individuals and diagnosed 64 new cases, a general new case detection rate (NCDR) of 13.4%; 60 were among 300 LSQ+ (NCDR-20%), while 4 were among 179 LSQ negative (NCDR-2.23%). In LG, Q7(65%), Q2(60%), Q1(45%), Q4(40%) and Q8(25%) were most frequent. All 2x2 crossings of these 5 questions showed a relative risk for leprosy ranging from 3 to 5.8 compared with NLG. All patients were multibacillary and presented hypochromatic macules with loss of sensation. LG anti-PGL-I titers were higher than NLG,

**Funding:** This work was supported by WHO Implementation Research Team of Ribeirão Preto Medical School; the Center of National Reference in Sanitary Dermatology focusing on Leprosy of Ribeirão Preto Clinical Hospital, Ribeirão Preto, São Paulo, Brazil; the Brazilian Health Ministry (MS/FAEPAFMRP-USP: 749145/ 2010 and 767202/ 2011); Fiocruz Ribeirão Preto - TED 163/2019 - Processo: N˚ 25380.102201/2019-62/ Projeto Fiotec: PRES-009-FIO-20. The funders had no role in study design, data collection and analysis, decision to publish, or preparation of the manuscript.

**Competing interests:** The authors have declared that no competing interests exist.

while 8.9% were positive for RLEP-PCR. The leprosy cases and anti-PGL-I spatial mappings demonstrated the disease spread across the town.

## Conclusions/Significance

Implementation actions, primarily LSQ administration focused on neurological symptoms, indicate hidden endemic leprosy in a nonendemic Brazilian state.

## Author summary

The prevalence of leprosy in the world and in Brazil is unknown. Although Brazil has effective disease notification systems, the data do not capture reality in the field, due to decreasing leprosy awareness, both in the community and among health professionals. Schools have decreased or stopped teaching about the disease, likely as a result of a massive campaign to eliminate leprosy as a public health problem around the world that focused almost exclusively on dermatological manifestations. The disease is primarily neural, affects mainly the reproductive-age population, and can generate disabilities leading to serious economic impacts on the individual and society. Despite modern diagnostic approaches, diagnosis of leprosy is still focused on clinical observation, given the scarcity of laboratory tests with good performance in terms of sensitivity and specificity. This makes leprosy diagnosis a challenge, especially for mild forms; *Mycobacterium leprae* can grow slowly and interact variably with the host, making leprosy as a highly complex disease. Only when the leprosy care policy in Jardinópolis municipality, São Paulo state inner, Brazil, was changed to hire a leprosy specialist for surveillance, was it possible to modify leprosy indicators revealing the hidden epidemic in the municipality. This study confirms hidden endemic leprosy in the municipality and informs implementation strategies for primary health teams using the Leprosy Suspicion Questionnaire (LSQ) on symptoms and signs of leprosy. LSQ is a low-cost, highly effective instrument to promote leprosy health education among community health agents and other health team workers, and among communities about the neurological symptoms that precede dermatological leprosy signs. This technique increases the likelihood of early diagnosis and treatment, avoiding disabilities, and consequently effectively halting disease transmission. The LSQ is an effective, low-cost screening tool for detecting new leprosy cases and increasing awareness of leprosy. The LSQ alerts community members and health professionals to detect even mild symptoms/signs of leprosy, a primarily neurological disease. Our data also demonstrate the importance of the leprosy specialist role to train and to supervise health teams to investigate incidence in communities that have long been established as nonendemic. The assumption of nonendemicity can and should change, due to the increase in the number of cases initially, and in view of the chronicity of leprosy and slow future decline. This change will in fact facilitate the much-desired elimination of leprosy as a global public health problem.

## Introduction

Leprosy is a chronic bacterial disease with a wide spectrum of clinical manifestations, caused mainly by *Mycobacterium leprae* [1,2]. This bacterium affects peripheral nerves and gives rise to deformities such as muscle wasting and wounds over anaesthetized areas of the body [3].

The long incubation period and insidious symptoms and signs of leprosy can make it difficult to diagnose and consequently delay treatment [1,4].

The World Health Organization (WHO) defined elimination of leprosy as a prevalence of below one case per 10,000 inhabitants, which was achieved for the world as a whole in 2000, and in most countries by 2005 [5]. The achievement of WHO's goal of eliminating leprosy was accompanied by a decrease in publicity about the disease and in teaching about leprosy in medical schools [6–8]. Despite effective leprosy treatment and massive public education efforts to facilitate leprosy control through the general health service in Brazil, clinical expertise (mainly among primary health care teams) regarding leprosy has declined [9,10], even among dermatologists. Implementation research is the scientific study of processes used in implementation of initiatives and factors that affect these processes, aiming to support and promote successful application of effective interventions [11]. Implementation research can be applied in the leprosy field to mitigate this hidden and neglected disease.

Although leprosy is still endemic in Brazil, indicators in São Paulo state indicate that the disease is controlled there. Bernardes-Filho *et al.* (2017) demonstrated unpreparedness among primary care teams to diagnose leprosy in Jardinópolis, São Paulo, following diagnosis of 24 new cases in 2015 (4.4 cases/10,000 inhabitants) [9], contrary to the presumed nonendemicity of São Paulo state since 2006.

Our objectives were to evaluate the effectiveness of the Leprosy Suspicion Questionnaire (LSQ) instrument to detect new leprosy cases, to confirm hidden endemic leprosy by clinicoimmunobiological evaluation and to establish clustering/mapping as a tool for identification of high-risk areas of leprosy, and to measure effectiveness of primary health worker training in Jardinópolis, São Paulo, Brazil.

## Subjects and methods

### Type of study

In this quasi-experimental study, the first part was a descriptive study of prevalence. The second part was an analytical, sectional study, relating leprosy-training interventions in human resources and LSQ effectiveness to changes in leprosy indicators.

### Ethics, consent and permissions

This study was approved by the Research Ethics Committee at the Clinics Hospital of Ribeirão Preto Medical School, University of São Paulo (protocol number 2.165.032, MH-Brazil). Written informed consent was obtained from every participant, including from the parent/guardian of each participant under 18 years of age. All procedures involving human subjects comply with the ethical standards of the Helsinki Declaration (1975/2008).

### Study area and population

This study was conducted March 2016–November 2019 in Jardinópolis, which had a population of 41,228 inhabitants. During the period 2010–2014, Jardinópolis had a mean leprosy new case detection rate (NCDR) of 4.1/100,000 inhabitants and prevalence of 0.73/10,000 inhabitants (IBGE 2017, SINAN 2017) [12,13]. For the leprosy active search action in Jardinópolis, meetings were held with the Primary Care and Epidemiological Surveillance Departments and the project team.

### Theoretical training of teams

In 2015, following employment of a leprologist in Jardinópolis, a high number of new leprosy patients were diagnosed, mainly in the northwestern census tracts of Jardinópolis [9].

| Leprosy Suspicion Questionnaire (LSQ)<br>Center of National Reference in Sanitary Dermatology<br>focusing on Leprosy<br>Ribeirão Preto Clinical Hospital, Brazil | |
|---|---|
| Name: | |
| Age: | |
| Check with "X" if there is some symptoms/signs present below | |
| 1. Do you feel numbness in your hands and/or feet? | |
| 2. Tingling (pricking)? | |
| 3. Anesthetized areas in the skin? | |
| 4. Spots on the skin? | |
| 5. Stinging sensation? | |
| 6. Nodules on the skin? | |
| 7. Pain in the nerves? | |
| 8. Swelling of hands and feet? | |
| 9. Swelling of face? | |
| 10. Weakness in hands? | |
| 11. Hard to button shirt? Wear glasses? Write? Hold pans? | |
| 12. Weakness in feet? Difficulty wearing sandals, slippers? | |
| 13. Loss of eyelashes? | |
| 14. Loss of eyebrows? | |

**Fig 1. Leprosy Suspicion Questionnaire (LSQ).**

In May 2016, theoretical leprosy trainings for community health agents (CHAs) and primary health care teams were performed. The LSQ was presented as a tool for active leprosy case detection. During the four weeks immediately following the training, the same LSQ that had been previously applied to a prison population [14] posing 14 simple questions about symptoms and signs associated with leprosy (Fig 1) was applied by CHAs during home visits. In the month following the return of the LSQ, the data extracted from the LSQ were transferred to an Excel version 6.0 spreadsheet.

## Practical training of teams

Respondents who reported some symptoms on the LSQ (LSQ+) and those who reported none (LSQ-) were invited to be evaluated clinically. During clinical care of these individuals, dermatologists experienced in leprosy explained leprosy concepts and performed dermatoneurological examinations. The primary heath teams participated in all consultations, which were

defined as week-long practical trainings that happened in August 2016, in May and August 2017. Subjects who consented to participate were referred for blood collection, and patients who received a leprosy diagnosis based on clinically well-established criteria as determined by at least two dermatologists were followed-up with a sample of a slit skin smear (SSS) for DNA (RLEP)-PCR. Following WHO's guidelines for Implementation Research in Health [11], in December 2017, new theoretical training to solidify the concepts of leprosy was carried out.

## Diagnostic criteria for leprosy

The subjects underwent a standardized clinical dermatoneurological exam according to Brazilian Ministry of Health guidelines. Leprosy diagnosis was made upon detection of at least one of the following signs/symptoms: a) lesion(s) and/or area(s) of the skin with changes in thermal and/or painful and/or tactile sensitivity; b) thickening of the peripheral nerve(s), associated with sensory and/or motor and/or autonomic changes; and/or c) presence of *M. leprae*, confirmed by intradermal smear microscopy or skin biopsy [15], which for this work was confirmed by RLEP-PCR. After certification by at least two experts, two groups were established: individuals diagnosed with leprosy (Leprosy Group, LG) and another with other individuals (Non-Leprosy Group, NLG).

## Assessment of anti-PGL-I titer by ELISA

Indirect ELISA was used to measure the anti-PGL-I IgM titer of all of the serum samples using the protocol previously reported [14,16]. The sample index was calculated by dividing their optical density (O.D.) per the established cut-off of 0.295; indexes above 1.0 were considered positive.

## DNA extraction and RLEP amplification

Total DNA extraction of earlobes and at least one elbow and/or lesion SSS sample using the QIAamp DNA Mini Kit (Qiagen, Germantown, MD, cat: 51306) was performed according to the manufacturers' protocol as described previously (2020) [14].

## Spatial epidemiology

The street addresses of all subjects included in this study were georeferenced with a handheld global positioning system (GPS) device (Garmin eTrex H, Olathe, KS, USA), or remotely geocoded using BatchGeo Pro (https://batchgeo.com). To produce maps of leprosy cases distribution in the town, we used QGIS 3.10.7-A Coruña (http://www.qgis.org). We drew point pattern maps for LG and NLG accounting for anti-PGL-I titration, as well as a Kernel density estimation map including a radius of 200 meters from each new case detected [9,17].

## Follow-up of patients

New leprosy cases diagnosed via active detection were followed-up. Evaluation of cases with suspected leprosy, referred by trained primary care professionals to the outpatient clinic at Epidemiological Surveillance, were performed only by the first author from September 2016 to November 2019.

## Statistical analysis

GraphPad Prism 8 software (GraphPad Software, San Diego, California, USA) was used to compare differences between both groups by the Wilcoxon-Mann-Whitney test, to measure LSQ effectiveness to predict diagnosis of leprosy, the Fisher's exact test, to calculate the relative

risk of the reported questions, and 2x2 contingency tables to cross the diagnosis (+/-) with LSQ (+/-), and to compare the number of reported questions 2x2. Differences were considered statistically significant at conventional levels with p<0.05.

## Results

During home visits by CHAs, 3,241 LSQ were applied throughout Jardinópolis. The CHAs had theoretical training, but no questions specifically mentioning leprosy were posed to respondents, who were instead asked to answer general questions about signs and/or symptoms and to return the LSQ to CHAs. A total of 1,054 (32.5%) LSQ were positive for one or more signs/symptoms (LSQ+) as described in Table 1. Among the LSQ+, each individual selected/reported 3.1 symptom/sign items on average; distributions by the number of selected answers in both groups are described in Table 1. Considering all individual LSQ+ respondents, the 5 most frequently reported signs/symptoms were Q2-Tingling (pricking)? (11.8%); Q4-Spots on the skin? (11.7%); Q7-Pain in the nerves? (11.6%); Q1-Do you feel numbness in your hands and/or feet? (10.7%) and Q8-Swelling of hands and feet? (8.5%).

Among 300 individual LSQ+ respondents evaluated clinically, 60 leprosy patients were diagnosed, a 20% NCDR. The most frequently reported sign/symptom items were Q7 (65%), Q2 (60%), Q1 (45%), Q4 (40%) and Q8 (25%) (Table 1). The mean number of reported items

**Table 1. Number of individuals ranked according to total signs and symptoms of leprosy marked on the LSQ in order of frequency (n = 1,054).**

| | Leprosy Suspicion Questionnaire (LSQ) | TOTAL (n) | % | NLG (n) | % | LG (n) | % | | |
|---|---|---|---|---|---|---|---|---|---|
| | Number of LSQ distributed | 3,241 | | - | - | - | - | | |
| | Number of LSQ with some mark (LSQ+) | 1,054 | 32.5 | | | | | | |
| | Number of LSQ respondees evaluated clinically | 479 | 14.8 | 415 | 86.6 | 64 | 13.4 | | |
| | Number of LSQ+ respondees evaluated clinically | 300 | 28.5 | 240 | 80 | 60 | 20 | | |
| | Number of LSQ- respondees evaluated clinically | 179 | | 175 | | 4 | | | |
| **Q** | **Symptoms and Signs (LSQ+)** | **TOTAL n = 1,054** | **%** | **NLG n = 240** | **%** | **LG n = 60** | **%** | **RR** | ***p*** |
| 1 | Do you feel numbness in your hands and/or feet? | 346 | 10.7 | 103 | 42.9 | 27 | 45 | 2.0 | <0.05 |
| 2 | Tingling (pricking)? | 384 | 11.8 | 111 | 46.2 | 36 | 60 | 2.9 | <0.05 |
| 3 | Anesthetized areas in the skin? | 87 | 2.7 | 31 | 12.9 | 11 | 18.3 | 2.2 | <0.05 |
| 4 | Spots on the skin? | 379 | 11.7 | 100 | 41.7 | 24 | 40 | 1.7 | <0.05 |
| 5 | Stinging sensation? | 177 | 5.5 | 35 | 14.6 | 13 | 21.7 | 2.3 | <0.05 |
| 6 | Nodules on the skin? | 145 | 4.5 | 36 | 15 | 3 | 5 | 0.6 | 0.28 |
| 7 | Pain in the nerves? | 375 | 11.6 | 89 | 37.1 | 39 | 65 | 4.3 | <0.05 |
| 8 | Swelling of hands and feet? | 276 | 8.5 | 65 | 27.1 | 15 | 25 | 1.5 | 0.12 |
| 9 | Swelling of face? | 72 | 2.2 | 28 | 11.7 | 6 | 10 | 1.4 | 0.45 |
| 10 | Weakness in hands? | 158 | 4.9 | 41 | 17.1 | 9 | 15 | 1.4 | 0.31 |
| 11 | Hard to button shirt? Wear glasses? Write? Hold pans? | 95 | 2.9 | 28 | 11.7 | 6 | 10 | 1.4 | 0.45 |
| 12 | Weakness in feet? Difficulty wearing sandals, slippers? | 115 | 3.5 | 34 | 14.2 | 10 | 16.7 | 1.8 | 0.55 |
| 13 | Loss of eyelashes? | 23 | 0.7 | 7 | 2.9 | 2 | 3.3 | 1.7 | 0.43 |
| 14 | Loss of eyebrows? | 27 | 0.8 | 8 | 3.3 | 2 | 3.3 | 1.5 | 0.53 |
| Total number of answers | | 919 | | 716 | | 203 | | | |
| Mean answers/individual | | 3.1 | | 3.0 | | 3.4 | | | |
| Min | | 1 | | 1 | | 1 | | | |
| Max | | 13 | | 11 | | 13 | | | |

Q: question number; n: number of checked questions; NLG: non-leprosy group; LG: leprosy group; LSQ+: Number of LSQ with some mark; LSQ-: Number of LSQ without any mark; RR: relative risk; *p*: p significance value.

**Table 2. Distribution of crossing frequencies between the five most marked questions of LSQ and respective risk relatives.**

| Q x Q | n | % | RR | *p* |
|-------|-----|------|-----|---------|
| Q2 x Q7 | 28 | 43.7 | 5.7 | p<0.05 |
| Q1 x Q2 | 23 | 35.9 | 3.0 | p<0.05 |
| Q1 x Q7 | 20 | 31.2 | 4.7 | p<0.05 |
| Q4 x Q7 | 15 | 23.4 | 5.8 | p<0.05 |
| Q2 x Q4 | 14 | 21.9 | 4.5 | p<0.05 |
| Q7 x Q8 | 11 | 17.2 | 3.6 | p<0.05 |
| Q1 x Q4 | 10 | 15.6 | 3.0 | p<0.05 |

Q: question number; n: number of leprosy patient; RR: relative risk; *p*: p significance value.

was $\geq$3 in both groups. Notable crossings among marked questions of new leprosy patients were Q2xQ7 (43.7%), Q1xQ2 (35.9%), Q1xQ7 (31.2%), Q4xQ7 (23.4%), Q2xQ4 (21.9%), Q7xQ8 (17.2%), Q1xQ4 (15.6%) (Table 2). All two-by-two crossings showed differences in relative risk (RR) for leprosy ranging from 3 to 5.8 compared with NLG. Among 179 LSQ-respondents, only 4 leprosy new cases were diagnosed (2.2% NCDR).

The probability of finding one new leprosy case among LSQ+ individuals was 15 times greater than among LSQ-, with an RR of 8.95 (95% CI 3.3–24.2, p<0.0001).

In total, 479 (14.8%) individuals were evaluated and 64 new cases were diagnosed, a 13.4% NCDR within the Jardinópolis population sample.

LG (n = 64) and NLG (n = 415) groups were established and characterized in Table 3.

Concerning the leprosy patients' clinical aspects, all were multibacillary, 100% presented dysesthesia in macular cutaneous areas (Fig 2), and 71.9% presented some peripheral nerve impairment; other aspects are presented in Table 4. In NLG, 335 (80.8%) had BCG vaccination scars, similar to LG (52; 81.3%) (Table 4).

**Table 3. Demographic characterization of Leprosy and Non-Leprosy groups.**

| Groups | TOTAL (n = 479) | | NLG (n = 415) | | LG (n = 64) | |
|--------|------|------|------|------|------|------|
| Sex | n | % | n | % | n | % |
| Male | 170 | 35.5 | 150 | 36.1 | 20 | 31.2 |
| Female | 309 | 64.5 | 265 | 63.9 | 44 | 68.8 |
| Age (years) | | | | | | |
| Mean | 43.6 | | 44.1 | | 40.5 | |
| Median | 47 | | 48 | | 37.1 | |
| Max | 94 | | 94 | | 77 | |
| Min | 2 | | 2 | | 7 | |
| Age range | | | | | | |
| < 15 | 58 | 12.1 | 53 | 12.8 | 5 | 7.8 |
| 15 |--- 20 | 29 | 6.1 | 24 | 5.8 | 5 | 7.8 |
| 20 |--- 30 | 50 | 10.4 | 44 | 10.6 | 6 | 9.4 |
| 30 |--- 40 | 56 | 11.7 | 42 | 10.1 | 14 | 21.9 |
| 40 |--- 50 | 62 | 12.9 | 52 | 12.5 | 10 | 15.6 |
| 50 |--- 60 | 94 | 19.6 | 79 | 19.0 | 15 | 23.4 |
| 60 |--- 70 | 86 | 18.0 | 78 | 18.8 | 8 | 12.5 |
| 70 |--- 80 | 35 | 7.3 | 34 | 8.2 | 1 | 1.6 |
| 80 |--- 90 | 8 | 1.7 | 8 | 1.9 | 0 | - |
| $\geq$ 90 | 1 | 0.2 | 1 | 0.2 | 0 | - |

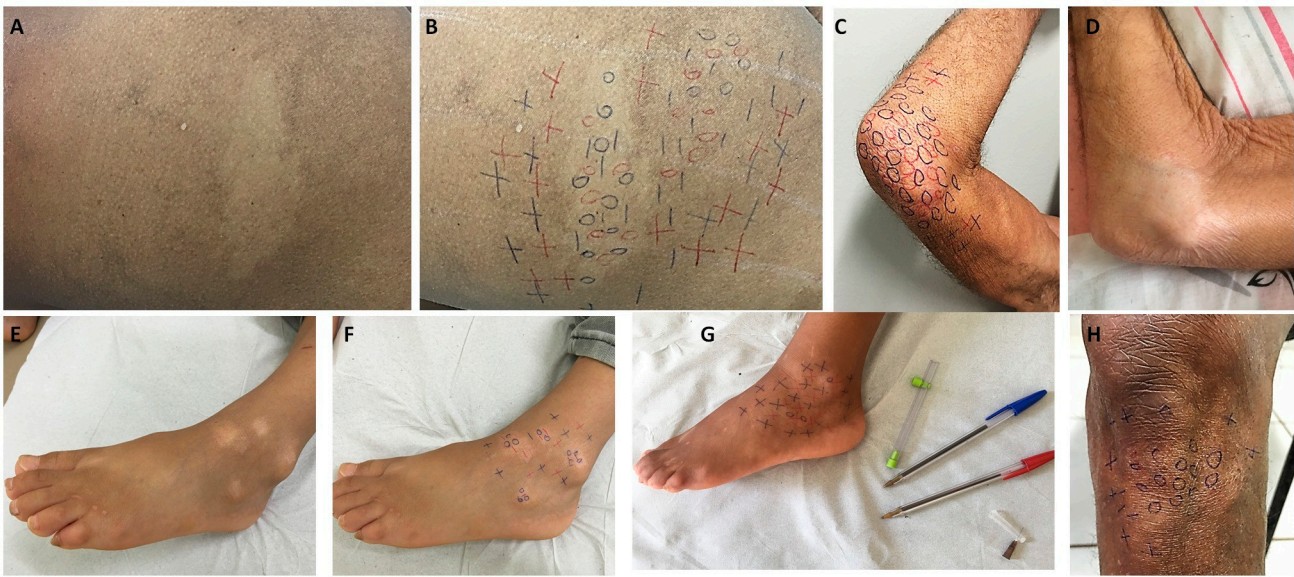

**Fig 2. Images of different locations (buttock, elbow, foot, and knee) of leprosy lesions.** (a) Hypochromatic macule on the left buttock; (b) the same patient, after performing the endogenous histamine test, below the initial macule observed, another hypochromatic, hypo-anesthetic macule became more evident due to the erythema surrounding the lesion; (c) hypochromatic anesthetic macule on the right elbow; (d) residual hypochromatic macule on the right elbow after 12 months of multi drug therapy; (e-f) hypochromatic hypo-anesthetic macules on the left foot; (g) the same previous patient with improvement of skin sensitivity with more normoesthesic points; (h) hypochromatic hypo-anesthetic macule on the left knee. Legend: 0 (anesthetic point); —(hypoesthetic point); + (normoesthesic point).

All individuals were tested for anti-PGL-I antibodies. Among 64 leprosy patients, 45 (70.3%) provided SSS (earlobes, elbows and/or skin lesion) for *M. leprae* DNA (RLEP) by PCR.

Considering anti-PGL-I antibody results, 40.6% were positive in LG (26/64) while 32.3% in NLG (134/415), an RR of 1.4 (95% CI 0.89–2.16, p = 0.19); LG presented higher OD titer mean than NLG mean (0.180) and p = 0.03. Anti-PGL-I indexes for both groups are shown in the Table 5. *M. leprae* DNA (RLEP)-PCR was performed in 45 patients from SSS; only 4/45 (8.9%) patients were positive.

Regarding spatial epidemiology (Fig 3), LSQs were distributed by CHAs all over Jardinópolis; 2,322 (71.6%) were distributed specifically in the northwest, a clustered region vulnerable to leprosy (Fig 3A), according our previous study [9]. Clinical evaluation involved individuals from all regions (Fig 3B). Leprosy cases were found in northwest, central and eastern areas (Fig 3C), although a hotspot of newly detected cases with the highest density clustered in the central northwest region (Fig 3D). The anti-PGL-I indexes in NLG were demonstrated in all regions, with high indexes around the LG cases (Fig 3E). In Fig 3F, one zoomed-in street view demonstrates a closed northwest region neighborhood with individuals with positive anti-PGL-I indexes around new cases in the same street.

Patients received clinical and therapeutic followed up with an experienced leprologist for 12 months, with reported clinical manifestations described in Table 6. There was notable improvement in neurological symptoms—skin sensitivity (82.5%), esthesiometry of feet (64.9%), and esthesiometry of hands (31.6%)—and in WHO impairment grading in 31.6% of patients. One hundred eighty-six household contacts were evaluated, and 8 new leprosy cases were diagnosed. During 3 years of work at the Epidemiological Surveillance Outpatient Clinic, one way to evaluate the effectiveness of implementation research after practical theoretical training, the primary care professionals referred 37 leprosy suspicious patients, and 11 (29.7%) new leprosy cases (Fig 4) were diagnosed.

**Table 4. Clinical characterization of Jardinópolis patients regarding the percentage of positivity to the clinical criteria used for the diagnosis of leprosy (n = 64).**

| Clinical features | LG (n = 64) | % |
|---|---|---|
| Dysesthesia hypochromic macular skin lesions | 64 | 100 |
| Localized irregular patches of circumscribed hair loss | 13 | 20.3 |
| Altered nerves on palpation (enlargement and/or pain and/or electric shock-like pain) | 46 | 71.9 |
| Endogenous histamine test performed | 32 | 100 |
| Incomplete | 32 | 100 |
| Esthesiometry of hands | 57 | 89.1 |
| Normal | 31 | 54.4 |
| Abnormal | 26 | 45.6 |
| Esthesiometry of feet | 57 | 89.1 |
| Normal | 8 | 14.0 |
| Abnormal | 49 | 86.0 |
| Leprosy classification | | |
| Borderline | 64 | 100 |
| WHO operational criteria | | |
| Multibacillary | 64 | 100 |
| WHO impairment grading | | |
| Grade 0 | 18 | 28.1 |
| Grade 1 | 30 | 46.9 |
| Grade 2 | 8 | 12.5 |
| Not evaluated | 8 | 12.5 |

| BCG scar | NLG | % | LG | % |
|---|---|---|---|---|
| 0 | 67 | 16.1 | 12 | 18.8 |
| 1 | 324 | 78.1 | 51 | 79.7 |
| ≥ 2 | 11 | 2.7 | 1 | 1.6 |
| Not evaluated | 13 | 3.1 | - | |

BCG: Bacillus Calmette–Guérin; LG: Leprosy group; WHO: World Health Organization.

Finally, because of implementation actions in Jardinópolis during our study, there were notable changes in leprosy epidemiologic rates, with significant increases after 2015 as described in Table 7.

## Discussion

Leprosy around the world has been neglected by many countries that do not report data to WHO [18]. Many studies describe active case detection for leprosy [19–21], but most are

**Table 5. Results of anti-PGL-I antibody measurements (anti-PGL-I index; cut off 0.295).**

| Groups | TOTAL (n = 479) | | NLG (n = 415) | | LG (n = 64) | |
|---|---|---|---|---|---|---|
| | n | % | n | % | n | % |
| Anti-PGL-I < 1 (negative) | 319 | 66.6 | 281 | 67.7 | 38 | 59.4 |
| Anti-PGL-I ≥ 1 (positive) | 160 | 33.4 | 134 | 32.3 | 26 | 40.6 |
| 1.0 \|--- 1.5 | 74 | 15.4 | 63 | 15.2 | 11 | 17.2 |
| 1.5 \|---2.0 | 42 | 8.8 | 35 | 8.4 | 7 | 10.9 |
| ≥ 2.0 | 44 | 9.2 | 36 | 8.7 | 8 | 12.5 |

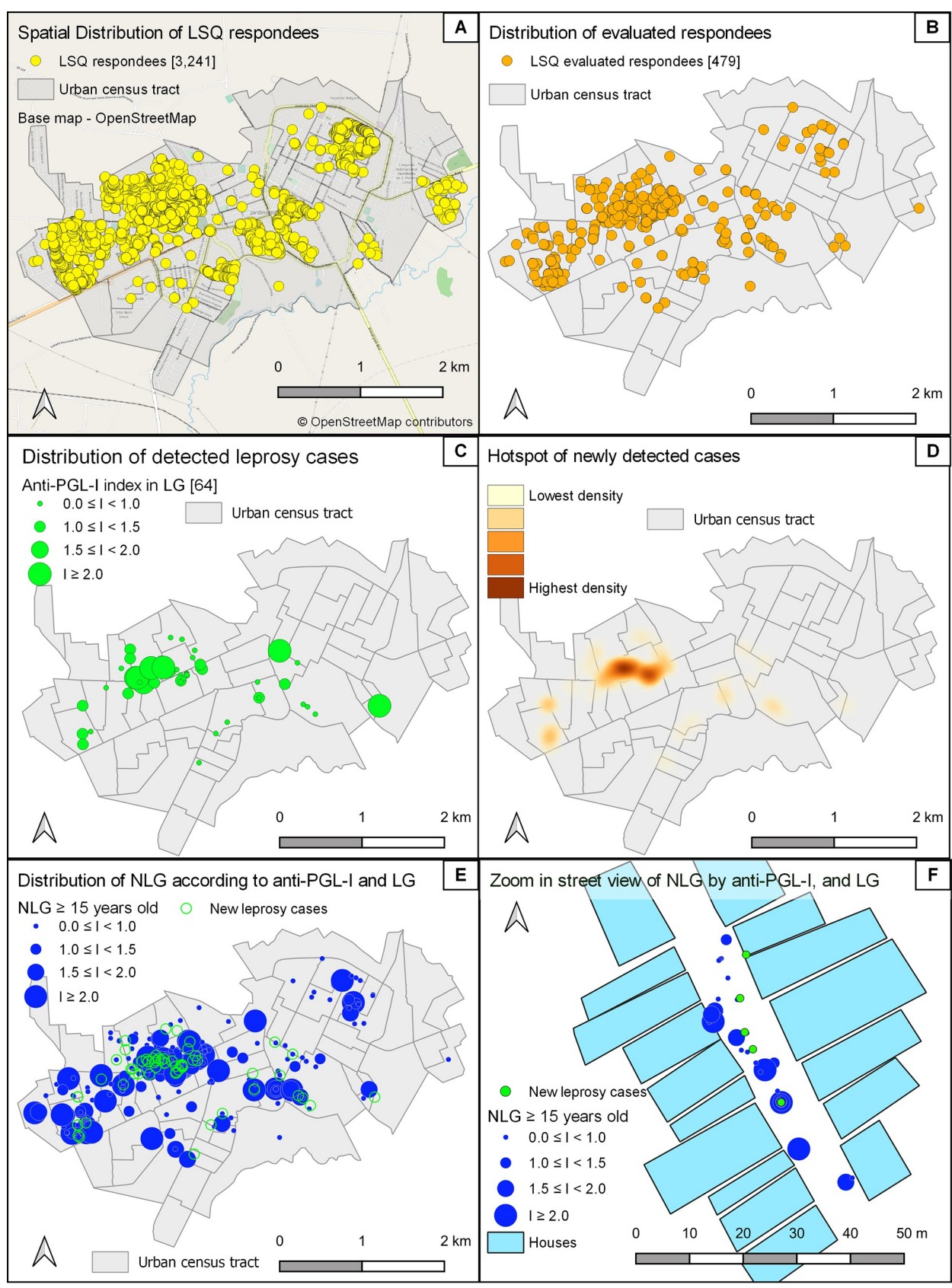

**Fig 3.** Spatial distribution of all LSQ applied specifically in the northwest (A), population sample of evaluated individuals (B), new cases of leprosy cases all over the municipality (C), hotspot of cases clustered in the center of northwest region (D), anti-PGL-I indexes in NLG in all regions with high indexes around the LG cases (E), and one zoon street view demonstrates the closed neighborhood with individuals with positive anti-PGL-I indexes around new cases in the same street of northwest region in the Jardinópolis (SP, Brazil) geographic area. www.openstreetmap.org.

applied by primary health care workers-based almost exclusively on skin signs, and suspicious cases are referred to specialists. Diagnoses are closed only after SSS and/or biopsy; neurological symptoms are unfortunately not considered, although there is no leprosy without nerve damage, as Fite stated in 1943 [22]. We aimed to educate primary health care providers on neurological as well as cutaneous signs/symptoms of leprosy but also implementing tools for the leprosy control program at Jardinópolis, a municipality with no endemic prevalence detected during 2010–2014.

Regarding the 5 most frequently selected signs/symptoms among all LSQ+ respondents, four are coincident with our work with prison populations: Q2(tingling/pricking), Q4(spots on the skin), Q7(pain in the nerves), and Q1(numbness in hands and/or feet) [14]. Among 60 new leprosy cases, Q4(spots on the skin) was the item with the lowest RR for disease, in contrast to neurological symptoms, namely Q7(pain in the nerves), Q2(tingling/pricking), Q5 (stinging sensation), Q3(anesthetized areas in the skin), and Q1(numbness in hands and/or feet), also coinciding with findings in the prison population [14].

Additionally, analyzing the 2x2 crossings, involvement of neurological symptoms such as Q7, Q2 and Q1 increased leprosy RR in relation to analysis of the questions individually. The greatest RR was reached associating neural pain with macular skin lesion (5.8).

When the LSQ was administered by CHAs, 60 patients were diagnosed among the 300 LSQ + individuals, a 20% NCDR, while 4 patients were diagnosed among 179 LSQ- (2.23% NCDR). This strategy showed a higher NCDR in the community than in a prison population [15] where the authors used the same LSQ, without CHA participation, detecting a 9.6% NCDR among LSQ+ respondents, while 1.83% among LSQ- [14].

Although we expected clustering in the northwest region, the leprosy NCDR in this area was 13.6%, similar to other regions (12.6%; p = 0.8), confirming the community has the same risk of leprosy, and the northwest region does not represent a natural clustered area. This was confirmed when we incorporated distribution by LSQ respondents and positivity to anti-PGL-I to calculate specific NCDRs, reaching 6.8% in northwest and 4.3% in other regions ($x^2$ = 2.2, p = 0.13) among LSQ+ respondents, and 40.5% and 38.2% respectively ($x^2$-Yates correction = 3.6, p = 0.06) among anti-PGL-I positives.

**Table 6. Clinical data and follow up in the leprosy patients treated.**

| Clinical evolution | LG (n = 57) | % |
|---|---|---|
| Improvement of skin sensitivity | 47 | 82.5 |
| Improvement of esthesiometry of hands | 18 | 31.6 |
| Worsening of hands' esthesiometry | 3 | 5.3 |
| Improvement of esthesiometry of feet | 37 | 64.9 |
| Worsening of feet' esthesiometry | 3 | 5.3 |
| Worsening of neurological symptoms (cramps, numbness and/or tingling) | 4 | 7.0 |
| Leprosy type 1 reaction | 1 | 1.7 |
| Neuritis | 5 | 8.8 |
| Dapsone-induced hemolytic anemia | 3 | 5.3 |
| Leprosy treatment dropout | 7 | 10.9 |
| Improvement of WHO impairment grading | 18 | 31.6 |

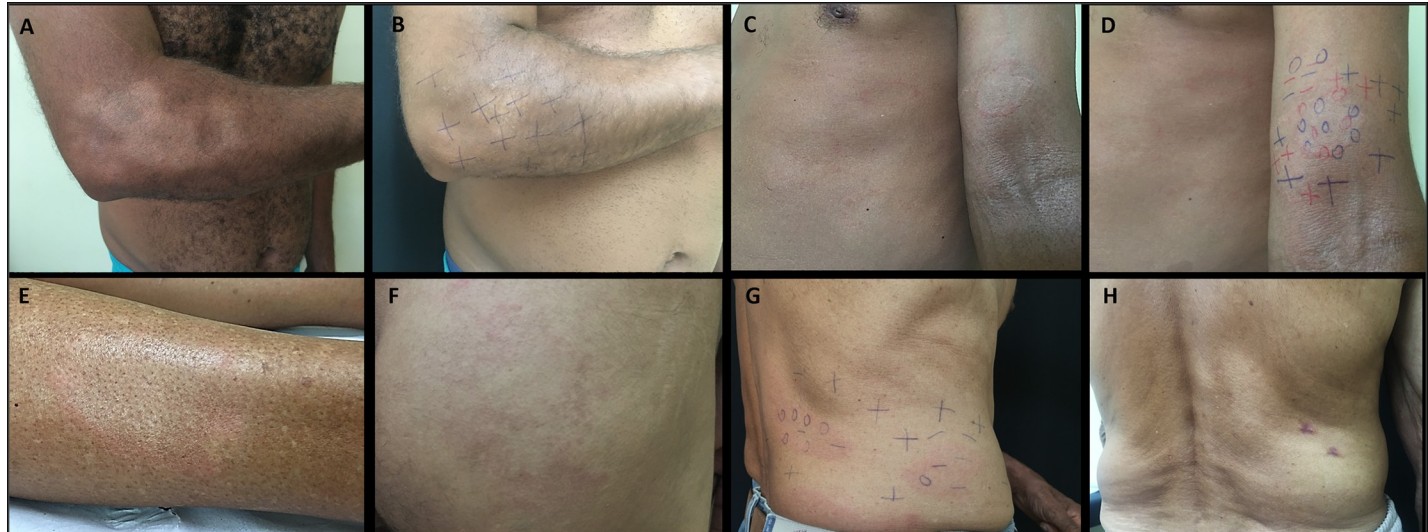

**Fig 4. Images of different locations (elbow, arm, leg, buttock, anterior and posterior trunk) of leprosy lesions.** (a) Hypochromatic anesthetic macule with irregular edges on the right elbow and forearm; (b) the same previous patient, after 12 months of multi drug therapy, with improvement of skin sensitivity; (c-d) typical borderline lesions on the left arm and trunk; (e) hypochromatic macule with infiltrated erythematous border on the right leg; (f) typical borderline lesions on the buttocks; (g) hypochromatic macules and erythematous plaques on the back; (h) the same previous patient, after 3 months of multi drug therapy with new hypochromatic macules on the back. Legend: 0 (anesthetic point);—(hypoesthetic point); + (normoesthesic point).

Regarding demographic characteristics, the number of female patients (44/60) was 2.2 times that of male patients (20/60), a contrast to Brazilian data with higher rates among males than females in 2014–2018 in all age groups [23]. This finding may result from LSQ administration in home visits during working hours; in Brazilian communities, women are more likely than men to be homemakers and thus be present at home. The distribution of patients by age group followed the same patterns as the Brazilian data, except diagnosis in five children under

**Table 7. Prevalence, new cases and new case detection rate (NCDR) in Jardinópolis from 2010 to 2019.**

| Year | Prevalence rate | Number of new cases | NCDR |
|---|---|---|---|
| 2010 | 1.06 | 4 | 5.31 |
| 2011 | 0.8 | 4 | 5.31 |
| 2012 | 0.52 | 0 | 0 |
| 2013 | 0.52 | 2 | 2.58 |
| 2014 | 0.74 | 3 | 7.38 |
| 2015 | 4.31 | 21 | 47.85 |
| 2016 | 23.61 | 96 | 226.64 |
| 2017 | 12.35 | 21 | 46.62 |
| 2018 | 2.80 | 14 | 18.65 |
| 2019 | 3.66 | 15 | 20.28 |
| Endemic pattern | Prevalence rate | Endemic pattern | NCDR |
| Low | < 1.0 | Low | 0 \|--- 2.0 |
| Medium | 1.0 \|--- 5.0 | Medium | 2.0 \|--- 10.0 |
| High | 5.0 \|--- 10.0 | High | 10.0 \|--- 20.0 |
| Very high | 10.0 \|--- 20.0 | Very high | 20.0 \|--- 40.0 |
| Hyper | ≥ 20 | Hyper | ≥ 40.0 |

NCDR: new case detection rate.

15 years of age (7.8%), this was higher than the national average of approximately 6%, further evidence for hidden endemicity.

In assessing signs and symptoms, all patients presented hypochromatic macules with loss of sensation, and the endogenous histamine test was incomplete in all those who were tested. Although macular presentations can be found in all forms of leprosy [24], the negative SSS is a common occurrence in 90–100% cases of macular leprosy [24], which reinforces that leprosy diagnosis is essentially clinical. At diagnosis, esthesiometry by Semmes-Weinstein monofilaments to assess tactile sensitivity was altered on the hands in 26/57 (45.6%) patients and on the feet in 49/57 (86%). Peripheral neural damage perceived on palpation was present in 46/64 (71.9%) of patients, coinciding with literature data that most macular leprosy patients display one or more enlarged nerves, indicating that macular cutaneous manifestations do not occur only in early leprosy cases [24]. Corroborating these data, approximately 60% of patients had some physical disability at diagnosis, 30 with grade 1 disability (46.9%) and eight with grade 2 disability (12.5%).

Regarding anti-PGL-I serology, 26 patients (40.6%) presented positive indices, and 8 (12.5%) with indices ≥2.0. Additionally, 134 (32.3%) individuals in NLG were positive, a very high rate in a state and region with officially controlled endemic disease. Brasil *et al.* demonstrated that a positive anti-PGL-I titer is a biomarker for *M. leprae* infection and carries around an 8-fold higher risk of disease progression [25]. In a prison population, the percentage of anti-PGL-I indices ≥2.0 among patients (35.3%) was higher than in NLG (4.5%) [14]. Although our present data demonstrate lower percentage of anti-PGL-I indices ≥2.0 among patients in the community than the penitentiary, probably due to cases with mild clinical signs, indices ≥2.0 among individuals without leprosy was almost double (8.7%). This demonstrates the actual risk of leprosy transmission in the community, and the importance of implementing actions to control leprosy, especially surveillance of contacts.

During follow-up of patients, following exclusive antimicrobial multidrug therapy (MDT), reported improvement in neurological symptoms like skin sensitivity was greater than 82%. Additionally, the peripheral nerve impairment was more consistent in the inferior limbs as demonstrated by a higher percentage of patients with altered feet esthesiometry at the diagnosis (86%), decreasing at discharge (21.1%), compared with lower values found in hands' esthesiometry, 45.6% at the diagnosis and 14.0% at discharge. These results differ from previous literature suggesting greater involvement of leprosy in upper limbs [26,27]. Notably, considering overall peripheral nerve involvement, WHO impairment grading improved in one-third of patients who received MDT.

Collection of follow-up data was possible because of the presence of a dermatologist experienced in leprosy working at leprosy reference outpatient clinic, improving medical assistance provided to patients in the municipality during this period, and facilitating leprosy training among primary care professionals.

The effectiveness of implementation strategies using the practical theoretical training, LSQ administration, and supervision by an expert may be measured additionally by the high number of new leprosy cases referred by the primary care team.

The use of participatory research processes to support learning and district health systems strengthening is a component of implementation research [28]. Empowering CHAs as first-line team players in the education of leprosy signs and symptoms for the community is extremely important. Further, involvement of primary care professionals in the early identification of individuals at risk for leprosy through the screening with LSQ has proven beneficial. Additional dissemination of the knowledge gained through our research is necessary to influence scale-up and widespread diffusion to other presumably nonendemic areas, not just endemic areas that are likely to have more readily detectable cases.

To definitely break the chain of leprosy transmission, health professionals need more education regarding early symptoms of leprosy and making a diagnosis incorporating neural dysfunction signs/symptoms rather than solely on dermato-morphological signs.

Until 2014, the year before the presence of a specialist at the head of Epidemiological Surveillance Outpatient Clinic, the town was officially nonendemic for leprosy. From 2015 onwards, Jardinópolis started to present high leprosy rates, standing out in the state of São Paulo, with indicators similar to that of high endemic and, even, hyperendemic regions [9]. Active leprosy case detection adopted in Jardinópolis as a public health policy raised official leprosy rates; this must be seen not just as a warning but also as an important change in management of the health care of the population.

The limitations of our study were the inability to evaluate all LSQ respondents and the non-reassessment of individuals with positive anti-PGL-I in the NLG.

## Conclusion

The LSQ proved to be an important screening tool for new leprosy cases by CHAs, providing a general detection rate of 13.4% among evaluated individuals and reaching 20% NCDR among the LSQ+ respondents—an 10-fold increase in risk of leprosy compared with LSQ- respondents. The crossing of questions 2x2, mainly involving neurological symptoms like neural pain (Q7), tingling/pricking (Q2), and numbness in hands and/or feet (Q1), despite spots on the skin (Q4) reinforces the importance of these symptoms for leprosy diagnosis. The LSQ also functioned as a consolidated, simple, cost-effective facility for health education, renewing awareness of leprosy signs and symptoms in the collective consciousnesses of the population and health professionals, and it should be recommended in the routine of screening and surveillance actions, complete or at least with the five most significant issues for the diagnosis of leprosy.

High anti-PGL-I indexes ($\geq 2$) could be a potential additional tool for leprosy screening in the community. The spatial epidemiology data disproved our categorization of the northwest region as a cluster; residence in all regions of the municipality presented the same risk of contracting leprosy.

Implementation research leveraging all these strategies for the active case detection for new leprosy cases at Jardinópolis revealed a hidden prevalence throughout the town. This represents both a warning to policymakers, and offers a lesson for the scientific community: municipalities will be only able to break chains of transmission and demonstrate they have controlled leprosy only through dedicated case detection with qualified professionals trained to recognize all clinical forms of leprosy, especially mild presentations, in concert with proper clinical and therapeutic follow-up of patients.

## Author Contributions

**Conceptualization:** Fred Bernardes Filho, Marco Andrey Cipriani Frade.

**Data curation:** Fred Bernardes Filho, Marco Andrey Cipriani Frade.

**Formal analysis:** Fred Bernardes Filho, Marco Andrey Cipriani Frade.

**Funding acquisition:** Marco Andrey Cipriani Frade.

**Investigation:** Fred Bernardes Filho, Claudia Maria Lincoln Silva, Glauber Voltan, Marcel Nani Leite, Ana Laura Rosifini Alves Rezende, Natália Aparecida de Paula, Josafá Gonçalves Barreto, Norma Tiraboschi Foss, Marco Andrey Cipriani Frade.

**Methodology:** Fred Bernardes Filho, Marco Andrey Cipriani Frade.

**Project administration:** Marco Andrey Cipriani Frade.

**Resources:** Fred Bernardes Filho, Marco Andrey Cipriani Frade.

**Supervision:** Marco Andrey Cipriani Frade.

**Validation:** Fred Bernardes Filho, Marco Andrey Cipriani Frade.

**Visualization:** Fred Bernardes Filho, Marco Andrey Cipriani Frade.

**Writing – original draft:** Fred Bernardes Filho, Marco Andrey Cipriani Frade.

**Writing – review & editing:** Fred Bernardes Filho, Claudia Maria Lincoln Silva, Glauber Voltan, Marcel Nani Leite, Ana Laura Rosifini Alves Rezende, Natália Aparecida de Paula, Josafá Gonçalves Barreto, Norma Tiraboschi Foss, Marco Andrey Cipriani Frade.

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
