## [Decision Letter · Decision Letter 0]

4 May 2021

Dear Dr. Frade,

Thank you very much for submitting your manuscript "Active search strategies, clinicoimmunobiological determinants and training for implementation research confirm hidden endemic leprosy in inner São Paulo, Brazil" for consideration at PLOS Neglected Tropical Diseases. As with all papers reviewed by the journal, your manuscript was reviewed by members of the editorial board and by several independent reviewers. The reviewers appreciated the attention to an important topic. Based on the reviews, we are likely to accept this manuscript for publication, providing that you modify the manuscript according to the review recommendations. Please pay careful attention to the very constructive criticisms and suggestions to improve the manuscript.

Sincerely,

Paul J. Converse

Associate Editor

Ana LTO Nascimento

Deputy Editor

Reviewer's Responses to Questions

**Key Review Criteria Required for Acceptance?**

**Methods**

-Are the objectives of the study clearly articulated with a clear testable hypothesis stated?

-Is the study design appropriate to address the stated objectives?

-Is the population clearly described and appropriate for the hypothesis being tested?

-Is the sample size sufficient to ensure adequate power to address the hypothesis being tested?

-Were correct statistical analysis used to support conclusions?

-Are there concerns about ethical or regulatory requirements being met?

Reviewer #1: The objectives are clear and the study is well-designed. The sample size and statistical analyses are well-fitted to the study. There are no ethical concerns.

Reviewer #2: The objectives of the study are clear and well presented.

Reviewer #3: 1.Abstract/Summary

Line: 38 "All patients were multibacillary and presented hypochromatic macules with loss of sensation"

**Results**

-Does the analysis presented match the analysis plan?

-Are the results clearly and completely presented?

-Are the figures (Tables, Images) of sufficient quality for clarity?

Reviewer #1: The results are clearly presented. My only concern is that the paper is quite long and some of the results could be regarded as superfluous, including the photos of skin lesions (Figures 2 and 5), and the follow-up data in Table 7 (this is a paper about improved case-finding). Table 5 shows that measuring anti-PGL-1 antibodies is of no value in establishing the diagnosis of leprosy, so Figure 3, which shows the same data, is not needed. In general, prevalence rates are no longer of much importance, while new case detection rates indicate more clearly what the epidemiological situation is. In this case, however, where the study is in a small town (pop ~ 40,000) actual case numbers should be given, rather than rates - this applies especially to Table 7, as the real catchment population is not known; people may have heard of better services being available and may have travelled from other surrounding towns. The mapping information is not very well analysed and is perhaps something of a distraction from the main message of the paper.

Reviewer #2: Results are properly presented

Reviewer #3: 1. line 197 "Although the CHAs had theoretical training, no questions specific mentioning leprosy were 198 posed to respondents, who were instead asked to answer general questions about signs and/or 199 symptoms and to return the LSQ to CHAs." 

But the title of the questionnaire is the "leprosy suspicion questionnaire" so I am not quite sure what you are signalling here? Are you trying to avoid anxiety amongst the responders about the diagnosis of leprosy? 

2. Esthesiometry is not a familiar term to me so I looked it up. It appears to mean the technique of measurement of sensation.

I think you mean "loss of sensation" or “anaesthesia” might these be more universally understood terms?

3. So why is the questionnaire not picking up paucibacillary patients. Is it obvious why? Perhaps you might include at some point in the manuscript your thoughts about this.

4. Would it be useful to indicate roughly how long the LSI took to complete? Ie the relative cost of including this within a future Public health screening approach?

**Conclusions**

-Are the conclusions supported by the data presented?

-Are the limitations of analysis clearly described?

-Do the authors discuss how these data can be helpful to advance our understanding of the topic under study?

-Is public health relevance addressed?

Reviewer #1: The main conclusions are that the Leprosy Suspicion Questionnaire is a valuable tool for active case-finding, as it brings into better focus the neurological aspects of leprosy, and that this helps in the training of health workers to be more aware of this component of the diagnosis of leprosy. This is an important message.

The authors mention two limitations of the study, but do not discuss any further implications for public health. The study was only able to evaluate 300 of the 1,054 LSQ+ respondents, for reasons that are not explained, but this is quite a serious flaw in the study. They also say that they were unable to reevaluate people who did not have leprosy but who were anti-PGL-1 positive. It seems to me that the LSQ is a very useful tool for case-finding, with high sensitivity, while testing for anti-PGL-1 antibodoes is shown in this study to be an extremely poor diagnostic test for leprosy.

It is surprising that the authors do not make any recommendations, or even suggest how they may plan to use the LSQ in future. For example, they have shown that five of the fourteen questions in the LSQ are more indicative of leprosy, so a reasonable approach may be to use a 'short' questionnaire, as part of an organized campaign in which all LSQ+ people are evaluated. This could be done in many different contexts, such as a skin camp, or a health education campaign, etc. Abandoning the use of anti-PGL-1 testing may free up resources to properly assess LSQ+ respondents.

Reviewer #2: Conclusions are in line with the paper

Reviewer #3: 1. 384 leprosy presence working at reference, I don’t understand this sentence. Can you reword?

2.Do you foresee this LSQ being used operationally as a screening tool by your public health teams? I thought this was what you intended but your focus at the end of the manuscript is of the benefit in involving CHA in research. I would have thought there is also potential benefit for case detection albeit one that will not identify all cases, and possibly not pauci bacillary disease.

**Editorial and Data Presentation Modifications?**

Reviewer #1: As mentioned above, the authors seem keen to present every last piece of data that they collected, sometimes to the detriment of the overall message. I think the questionnaire they have developed is a very important advance for case-finding.

Also Ref 14, which is said to be in press, is now available: : Bernardes Filho F, Santana JM, de Almeida RCP, Voltan G, de Paula NA, Leite MN, et al. (2020) Leprosy in a prison population: A new active search strategy and a prospective clinical

analysis. PLoS Negl Trop Dis 14(12): e0008917. https://doi.org/10.1371/journal.pntd.0008917

Reviewer #2: only more information and discussion on SSS

Reviewer #3: (No Response)

**Summary and General Comments**

Reviewer #1: There is nothing specifically wrong here, but much of the material presented is irrelevant to the main message of the paper, which is the value of a questionnaire focusing on neurological symptoms in combination with training of health staff on this aspect of leprosy diagnosis. The title of the paper could be more eye-catching, such as "The Leprosy Symptom Questionnaire (LSQ) focuses on common neurological sypmtoms of leprosy and is a highly effective case-finding tool."

Reviewer #2: Dear Authors,

I read your paper with interest having read the previous paper about this town and the use of the questionnaire in a prison population. I am impressed about the results the questionnaire gave. 

It was a particularly good choice to use neurological symptoms, because there is no leprosy without nerve involvement.( Fite GL. 1943. Leprosy from histopathologic point of view. Arch Pathol Lab Med 35:611–644.)

The paper on the questionnaire contains at the end a second part on the spatial distribution. I can not find this in the abstract.

I have a few questions and remarks: 

79 deformities are muscle wasting contractures and absorption, I doubt injuries.

87 even under dermatologists

Training not trainings

146 I assume that SSS was used for diagnosis with AFB staining.

157 I see AFB was done.

167 positive PGL1 does not mean infection, just contact. Thus, better use this instead of subclinical.

206 How where the positive responders selected for clinical evaluation?

209 you mention both groups the LG and NLG from the 300?

212 How were the non-responders selected for clinical evaluation.

In table 4 I expected positive SSS for AFB. But there was none?

259 How many had a positive AFB when, so few had a positive PCR? What I do not understand , did you not find a positive SSS? And you had 64 MB patients, did you classify only on the number of patches.? How many patches had no sign of nerve damage? In LLs and may be BL, I would expect only the early patches with detectable damage. You did it very carefully. In a borderline group you usually find patches with loss. But not all. How many did you test with no detectable nerve damage? 

I liked the street view with Leprosy patients and PGL1 positive contacts.

The primary healthworkers seemed well trained!

316 from here I understand again that diagnosis was confirmed after SSS. Like you mentioned in the introduction why can I nowhere find the result? My fault? For me 64 MB patients and no positive SSS is strange. 

345 Good to mention here that among the women 11% was positive and among the male 12.3% So the man/women may be in line with the country’s findings.

353 Here I find a mention of SSS. How would you classify your patients?. Ridley Jopling BT or old Brazilian Inditerminate? Could you check staining of your SSS AFB. Because M.leprae is only weakly acid fast. May be that count for your negatives. At your institite (in Ribeiroa Preto) it will be done properly but where were the SSS stained? In a TB lab? I do not doubt that 100% of the clinical diagnosed patients were leprosy patients. But in my experience not 80-100% of macular leprosy in Brazil are SSS negative.

Reviewer #3: Thank-you for the opportunity to review this interesting approach to the use of risk questionnaires for disease identification by CHA. I hope the comments are useful.

PLOS authors have the option to publish the peer review history of their article (what does this mean?). If published, this will include your full peer review and any attached files.

Reviewer #1: No

Reviewer #2: No

Reviewer #3: Yes: Dr Claire Fuller

Figure Files:

Data Requirements:

Reproducibility:

References

---

## [Editor Report · Decision Letter 1]

14 May 2021

Dear Dr. Frade,

Thank you very much for submitting your manuscript "Active search strategies, clinicoimmunobiological determinants and training for implementation research confirm hidden endemic leprosy in inner São Paulo, Brazil" for consideration at PLOS Neglected Tropical Diseases. As with all papers reviewed by the journal, your manuscript was reviewed by members of the editorial board and by several independent reviewers. The reviewers appreciated the attention to an important topic. Based on the reviews, we are likely to accept this manuscript for publication, providing that you modify the manuscript according to the review recommendations. 

The revisions have largely reflected the suggestions of the reviewers. However, there were a few instances, indicated in comments and corrections in the attached Word document, that should be made to fully answer the reviewer's concerns. Ideally, the manuscript would be carefully edited throughout for English grammar, usage, and clarity. Regarding the title, the point made by Reviewer 1 is highly pertinent. There is not a lot of immunobiology described here. You might consider: "A questionnaire-based search strategy for clinical determinants and training for implementation research confirms hidden endemic leprosy in inner Sāo Paulo, Brazil"

Sincerely,

Paul J. Converse

Associate Editor

Ana LTO Nascimento

Deputy Editor

Figure Files:

Data Requirements:

Reproducibility:

References

---

## [Editor Report · Decision Letter 2]

20 May 2021

Dear Dr. Frade,

We are pleased to inform you that your manuscript 'Active search strategies, clinicoimmunobiological determinants and training for implementation research confirm hidden endemic leprosy in inner São Paulo, Brazil' has been provisionally accepted for publication in PLOS Neglected Tropical Diseases.

Best regards,

Paul J. Converse

Associate Editor

Ana LTO Nascimento

Deputy Editor

---

## [Editor Report · Acceptance letter]

9 Jun 2021

Dear Dr Frade,

We are delighted to inform you that your manuscript, "Active search strategies, clinicoimmunobiological determinants and training for implementation research confirm hidden endemic leprosy in inner São Paulo, Brazil," has been formally accepted for publication in PLOS Neglected Tropical Diseases.

Best regards,

Shaden Kamhawi

co-Editor-in-Chief

Paul Brindley

co-Editor-in-Chief
